# The Effect of Microparticles on the Storage Modulus and Durability Behavior of Magnetorheological Elastomer

**DOI:** 10.3390/mi12080948

**Published:** 2021-08-11

**Authors:** Mohd Aidy Faizal Johari, Saiful Amri Mazlan, Nur Azmah Nordin, U Ubaidillah, Siti Aishah Abdul Aziz, Nurhazimah Nazmi, Norhasnidawani Johari, Seung-Bok Choi

**Affiliations:** 1Engineering Materials & Structures (eMast) iKhoza, Malaysia-Japan International Institute of Technology (MJIIT), Universiti Teknologi Malaysia, Kuala Lumpur 54100, Malaysia; mohdaidyfaizal@graduate.utm.my (M.A.F.J.); nurazmah.nordin@utm.my (N.A.N.); aishah118@gmail.com (S.A.A.A.); nurhazimah@utm.my (N.N.); norhasnidawani@utm.my (N.J.); 2Mechanical Engineering Department, Faculty of Engineering, Universitas Sebelas Maret, J1. Ir. Sutami 36A, Surakarta 57126, Indonesia; ubaidillah_ft@staff.uns.ac.id; 3Department of Mechanical Engineering, The State University of New York, Korea (SUNY Korea), 119 Songdo Moonhwa-ro, Yeonsu-gu, Incheon 21985, Korea

**Keywords:** anisotropic, durability, isotropic, linear viscoelastic region, magnetorheological elastomer, microparticles, storage modulus

## Abstract

This paper presents the effect of the micro-sized particles on the storage modulus and durability characteristics of magnetorheological elastomers (MREs). The initial phase of the investigation is to determine any associations among the microparticles’ weight percent fraction (wt%), structure arrangement, and the storage modulus of MRE samples. In order to carry out this, both isotropic and anisotropic types of MRE samples consisting of the silicone rubber matrix and 50, 60, 70, 75, and 80 wt% microparticles of carbonyl iron fractions are prepared. It is identified from the magneto-rheometer that the increase in storage modulus and decrease in linear viscoelastic region limit are observed in varying consistency depending on wt% and particle arrangement. The consistency of this dependency feature is highlighted by superimposing all of the graphs plotted to create the proposed the samples’ behavior model. In response to increasing magnetic stimulation, a sample of 70 wt% microparticles with an isotropic arrangement is found to be significant and stable. The experimentally defined fraction is then used for the durability test as the second phase of the investigation. During this phase, the durability evaluation is subjected to stress relaxation for an extended period of time. After undergoing durability testing, storage modulus performance is decreased by 0.7–13% at various magnetic stimulation levels. This result directly indicates that the storage modulus characteristics of different forms of MRE are sensitive to the different iron particle fractions’ and microparticles’ alignment. Therefore, important treatments to alter the storage modulus can be undertaken before the practical implementation to accommodate any desired performance of MRE itself and MRE application systems.

## 1. Introduction

Magnetorheological elastomers (MREs) are elastomeric materials that are uniquely receptive to external magnetic stimuli because they are composed of magnetically permeable microparticles structured inside a polymer matrix [1]. MRE, as a novel smart material, has modifiable mechanical properties as a result of the interaction of magnetizable microparticles embedded in a non-magnetic matrix. In recent years, MRE has emerged as a potential solution to many material technology developments, similar to magnetorheological fluid (MRF), but in solid form. MRE materials, which are responsive to magnetic stimuli and have variable stiffness, are gaining popularity in a wide range of critical applications, including actuators [2], transport, and other structural industries. Because of the advantages of MRE, it has also become a popular material choice for advanced engineering applications. These findings highlight the importance of considering some key technological issues ahead of time. MRE is increasingly being used in critical applications such as electronics and sensing components [3], microfluidics [4], military [5], radio-absorbing devices [5], and a variety of other promising fields. MRE technology and growth in this field represent a significant step forward in the never-ending search for material optimization, as material is the final frontier of advancement. A successful curing process in MRE fabrication yields outstanding properties to meet requirements for designing a certain typical specimen geometry in testing and analysis [6,7]. The matrix has a significant impact on the properties of MRE and an appropriate matrix form and fraction should be used to induce storage modulus [8,9]. According to a review on MRE durability evaluation [10], the usual matrix used for MRE evaluation may be categorized into three types: natural rubber (NR), synthetic rubber (SR), or a combination of NR and SR. MRE is classified into isotropic and anisotropic. In isotropic MRE, the magnetic particles are uniformly dispersed in the matrix without external magnetic influence and, contrary to anisotropic MRE, the magnetic microparticles form chain-like arrangements in the matrix from strong external magnetic fields applied during the curing process [11,12]. Varying or changing process parameters during the fabrication process can easily manipulate or affect the material properties of the MRE. Therefore, special attention must be given to the fabrication process in producing MRE samples, so that the full potential of MRE is readily achievable. Furthermore, the properties of the embedded microparticles, such as size, shape, and fraction to matrix material, have a significant impact on MRE magnetic responsiveness [13]. Aside from these particle parameters, a successful study [14] established that particle surface characteristics, such as coating, play an important role in tuning the MREs’ properties. Related studies on the effect of particle fraction and arrangement of MRE were reported by previous researchers [15,16,17,18,19,20,21]. Previous research [12,22,23,24] discovered that preparing anisotropic samples by varying the spatial distribution of magnetic microparticles in the elastic matrix increased the MR effect, which warrants further experimental and theoretical study.

The microstructure, especially the size and arrangement of the microparticles, has been shown to have a strong influence on the rheological properties and performance of MRE in previous research [7,25,26,27]. Garcia-Gonzales et al. [7] made the most important recent advances in this field by designing and formulating an MRE nonlinear constitutive model that accounts for these dependencies. Furthermore, Winger et al. [13] found evidence that the microparticles fraction plays a critical role and could have a significant impact on the magnetorheological (MR) effect. Similarly, Lokander and Stenberg [28] and Khayam et al. [29] previously discussed the particle fraction aspect by introducing critical particle volume concentration (CPVC) in relation to the increase of the MR effect and particle alignment ability. Heretofore, MRE has attarcted the attention of researchers because of its ability to exhibit variable stiffness and damping properties when exposed to an external magnetic field. As a result, MRE is now used in a variety of engineering applications [30]. As a result, in addition to selecting the appropriate alignment and particle fraction to obtain the best potential candidates for rheological characteristics, understanding the selected sample for durability performance is quite fascinating and contributes to the available literature. Motivated by this prospect, this study offers a significant opportunity to advance the understanding of MRE microscopic durability performance by stress relaxation. The existing and current literature on MRE durability focuses on mechanical and rheological properties in particular. The properties’ contribution to MRE degradation performance is largely unknown. There has been no in-depth study of the interaction of these properties with the physical vicissitudes of the MRE. The specific contribution of this work is to present a new microstructural consideration that can make the durability determination in the performance control of MRE structure more complete. To carry out this, attempts have been made to link strain localization caused by stress relaxation to MRE microstructure and its relationship to storage modulus reduction. The current study also adds to the existing body of evidence and contributes to the understanding of how MRE can adjust the storage modulus to compensate for a decrease in performance under durability conditions. The contribution from these results demonstrates MREs’ superior ability to adapt to durability cause of storage modulus depletion and improve to desired quality using magnetic stimuli.

## 2. Materials and Methods

### 2.1. MRE Fabrication

The isotropic and anisotropic arrangements of MRE sample-based silicone rubber were fabricated by incorporating 50%, 60%, 70%, 75%, and 80% weight fraction microparticles of soft carbonyl iron particles (CIPs) (d50 = 3.8–5.3 µm, CC grade, supplied by BASF, Germany) into room temperature vulcanized (RTV) silicone rubber supplied by Nippon Steel Co., Tokyo, Japan at 25 °C. The mixtures were cured by adding a curing agent (≈0.1 wt%) and allowed to solidify for 2 h. The curing agent, NS625B (Nippon Steel Co., Tokyo, Japan), was used as a cross-linking agent and the best amorphous MRE matrix characteristic was determined. The curing pressure was uniformly 12.963 kPa throughout the process and there was no evidence of gravity segregation. The curing procedure was carried out in a cylindrical mold with a diameter of 50 mm and thickness of 1.2 mm of a circular sinking section inside the mold at off-state condition for isotropic MRE and under a magnetic field of 0.3 T (parallel to the thickness) for anisotropic MRE. For anisotropic MRE, the mold was placed in a magnetic bobbin with a known magnetic polarization vector. Because the material inside the mold had not yet solidified, the matrix was viscous enough for the CIP to move freely and align themselves with the field lines, as shown in Figure 1. Magnetic polarization has an effective time of a few seconds, according to estimates. The sample was kept in the mold inside the bobbin for about 30 min, until it was completely solid and had a permanent structure. As a result of aligning the CIP in a matrix in the form of the magnetic field, an MRE with directed (anisotropic) structure was obtained [20]. Finally, an MRE circular disc sample was cut out of the original prepared MRE disc sheet using hollow hole punch tools to a diameter of 20 mm and a nominal thickness of 1.2 mm for rheological analysis.

### 2.2. Rheological Testing

The rheological behavior of the MRE sample was tested under torsional shear mode and measured by an oscillation parallel plate rheometer (Physica MCR 302, Anton Paar Company, Graz, Austria). The rheometer was priorly initialized to desired test condition (temperature, force, and gap) and the measuring tool (rotary disc parallel plate, 20 mm diameter) was aligned and attached to the quick connector coupling. The sample was centrally placed on the stationary base mount of the rheometer and preloaded before the oscillation using rotary disc pp20 rod and then subjected to normal force during the test to avoid wall slip [31].

The measurement of the significant viscoelastic property was conducted at a controlled ambient temperature of 25 °C by utilizing Viscotherm VT2, Anton Paar (Graz, Austria). Magnetorheological device MRD 70/1T was used to generate a homogeneous magnetic field perpendicular to the sample. The magnetic fluxes produced by the test range from 0 to 850 mT when a current of 0 to 5 A was applied [32]. However, flux density generated across the sample at a similar current applied was different relying upon CIP fraction. At steady-state condition, the shear amplitude was set at a linear ramp of dynamic strain sweep from 0.0001% to a maximum of 10%. The angular velocity was constantly set at 1 rad/s. Each sample was initially tested at zero magnetic field and a similar procedure was repeatedly performed for increasing magnetic field. Each interval at the start of the magnetic application underwent magnetic degaussed to ensure zero interference at the beginning of the repeated procedure. The storage modulus characteristic varied discretely throughout the oscillatory test at increasing magnetic fields ranging up to 5 A, with 1 A of current interval. For the assessment after the durability condition, a similar approach was applied. Figure 2 illustrates the rheometer (MCR302) and sample condition in the magnetorheological device MRD 70/1T.

### 2.3. Stress Relaxation Durability Test

Stress relaxation was used to analyze the long-term durability behavior prediction from time-dependent viscoelastic properties of MRE. Under a constant deformation condition, stress worsened with time during stress relaxation. The shear stress relaxation behavior of the MRE samples were evaluated using an oscillation parallel plate rheometer in torsional shear mode, as with previous rheological evaluations. The sample was subjected to a 5.9 N applied initial force during the pre-compression process, while the compressive strain remained constant. The shear deformation, on the other hand, was held constant at 0.01% throughout the test, which was highlighted as an attempt to stress relaxation of MRE samples closest to their resting state. The 0.01% strain value was obtained from the rheological results of all samples using linear viscoelastic (LVE) parameters and was measured using the visual method and the procedure defined in the literature [18,33,34]. The test frequency was kept constant at 1 Hz to represent the actual working conditions in the application and to account for the effects of the shear velocity gradient during the test. The time interval for each test condition was set at every 2000 s for the starting condition and gradually increased to 4000 s for each interval until a significant total test period of 84,000 s, which translates to almost 23 h, was obtained. This practice has expanded the scope of behavior observation.

## 3. Results and Discussion

Figure 3 depicts field emission electron microscope (FESEM) images of particle dispersion at high (70 wt%) and low (50 wt%) fractions. For an isotropic sample, the distribution on the high wt% sample in Figure 3a has more pultruded CIP and uniform distributions. As the fractions increased, the distribution flow during the curing process was much slower than in the lower fraction sample, indicating less desirable distribution of CIP. As a result, uniformly distributed and imperceptible pultrusion of CIP was observed in the lower fraction sample, as shown in Figure 3c. An anisotropic arrangement sample demonstrates more aligned CIP for a high wt% sample, as shown in Figure 3b. Because the CIPs are in closer contact with each other, the interaction between them is more efficient during magnetic stimulation. The magnetic energy was converted into kinetic energy, which allowed for more interaction between CIP and allowed it to be easily aligned with the magnetic field direction. However, at a lower wt% fraction, when the sample is in a condition with a large gap between the CIP, the interaction becomes much less and is difficult to align as magnetic stimulation is applied. As a result, less systematic particle alignment can be seen, as shown in Figure 3d.

MRE at different microparticle ratios and arrangements has shown significant characteristics and behavior of an increasing trend in storage modulus, as shown in Figure 4. This attribute corresponds to the increasing flux density applied to the MRE sample during the test. Theoretically, magnetic flux density has stimulated the carbonyl iron particles, which has high permeability, high saturated magnetization, and low remnant. These characteristics very much influence the MRE behavior in maintaining magnetic domains. However, the fraction and distribution of particles would change the performance of MRE. Figure 4 shows that MRE samples with anisotropic arrangements at similar microparticle of CIP fractions to isotropic MRE perform better in terms of deformation energy absorption, as evidenced by higher shear storage modulus values. Entire tests exhibited a similar improving pattern of storage modulus, although it was saturated at higher flux density stimulation. The rheological response for both arrangements showed that the storage modulus changes dramatically, particularly at low magnetic stimulus applications. As magnetic use rose, the change in storage modulus values was rather minimal. Saturation of storage modulus at a higher magnetic field denoted the sample to have a more solid-like property and indirectly improved strength and rigidity. An apparent increase of storage modulus could be observed at 1 to 3 A and a slighter one between 4 and 5 A. CIP reached its maximum level magnetization at a higher applied current (>3 A), so the additional current increase had no effect on its performance. As anisotropic MRE has a chain-like structure, it exhibited more pronounced field-induced modulus changes than the isotropic distribution.

Nonetheless, according to the findings of Qiao et al. [35], isotropic arrangement strengthened with increasing applied current, while retaining their durability and elasticity. Furthermore, as shown by the work of Gong et al. [36], isotropic MREs have a high potential for generating significant storage modulus improvements when magnetic stimuli are applied. These claims were then proved in this work, which showed that 70 wt% of isotropic samples performed consistently under the influence of a magnetic field, with a stable improvement in the ability to store deformation energy elastically. Their result was also consistent with previous work of Davis [37] on particle fraction optimization. The study stated that the best fraction of particle volume for the highest fractional shift in modulus at saturation was estimated to be 27 vol.%. According to the numerical prediction and experimental evidence, the best possible fraction was 70 wt%, which equals 28 vol.%. This was also the reason most of the researchers [38,39] in this sort of investigation preferred 70 wt% CIP fractions. The previous comprehensive review [10] provided a detailed discussion of the microparticles wt% chosen by researchers, especially for the durability evaluation.

Disregarding the influence of particle size, the microparticles’ fraction in the MRE sample plays an important role and has an effect on their elasticity and rigidity. A higher fraction CIP decreased the amount of matrix carrier and critical concentration of the microparticles. The response of MRE sample with higher fraction to the magnetic field seems better at a higher CIP fraction, nevertheless compelled to tolerate with their stiffness, brittleness, and elasticity, which is represented by LVE. As shown in Figure 4a–d, the LVE region is much shorter and shows a sudden downturn compared with the plotted graph in Figure 4e–h. Thus, the shorter nature and sudden downturn of 75 and 80 wt% CIP indicated brittle behavior and less elasticity. However, the elastic portion from the LVE region was much longer for 50, 60, and 70 wt% CIP. The gradual downturn of these fractions stipulates less brittle behavior. In addition, at a higher wt% fraction of CIP, escalating initial value of storage modulus for applied current has a substantial range compared with at a lower wt% fraction.

In this research, phenomena of the Payne effect were also established and similar trends were previously reported by Sorokin et al. [40] and other researchers [41,42,43,44] for CIP-filled MRE. The Payne effect occurred with a sudden decrease in storage value under a condition with small strain deformation and saturating at rather large deformations. In this case, storage modulus decreased rapidly with increasing strain, as shown in Figure 4h at 0.001% strain. Apparent Payne effect behavior was observed at a higher magnetic field (2–5 A). Moving beyond sufficient larger strain deformation, storage modulus approaches a lower bound deliberately. As the Payne effect is associated with the phenomenon of softening stress, only the MRE sample at 60 wt% experienced the event. The Payne effect occurred as the viscoelastic storage modulus’s subserviency to strain amplitude. It is associated with changes in the microstructure of the material caused by deformation [45]. At anisotropic arrangement, microparticles are permitted to undergo situations where there are more gaps because the polymer cannot fit all the gaps while distributed into huge matrix fractions during the curing process [17]. The breakdown of structured particle alignment can account for this phenomenon [46]. The fragment of the particle’s structure organization was then confined with an insignificant amount of matrix, and was thus less matrix dependent, less elastic, and exhibited brittle failure behavior.

The primary motivation for measuring the energy stored in the MRE due to elastic deformation was storage modulus (G′) behavior. The establishment of storage modulus characteristic for entire particle fraction and arrangement was designated to the model of MRE behavior, as shown in Figure 5. This model is the superimposed version of the established results plotted previously and describes the effect of particle fraction on the performance of MRE at various magnetic flux applications. Higher microparticles’ fraction leads to better stiffness, but compromises the elastic potentiality. Moreover, an increasing particle content decreased the LVE region and shortened the linearity limits. As a result, the range in which the test can be carried out without destroying the structure of the sample was greatly influenced.

The obtained storage module was used to determine the MRE’s energy characteristic due to elastic deformation. For both types of distributions, the storage modulus behavior in samples with a high wt% of CIP exhibits significantly higher G′ than samples with a low wt% of CIP. Figure 5 depicts this with a dotted line graph. The initial G′ and its range of increment towards magnetic stimulation are the obvious differences in G′ observed in the proposed model. In terms of molecular scale durability, stress relaxation could occur and be involved by changing the structure of the molecular chain. In particular, stress relaxation phenomena in MRE have theoretically occurred via a variety of mechanisms. These include cross-link disengagement, elastic stretching, inelastic deformation, structural shift by phase transformation, microphase separation, and microplasticity. Durability, which is caused by stress relaxation, maintains the structure in a strained state for a finite interval of time, resulting in some amount of plastic strain. This plastic strain was only found in a very narrow region known as shear bands. The final stage will be the formation of shear bands as a result of localized strain. The process suffered the amorphous matrix and allows the molecular structure to undergo permanent deformation, a non-reversible change in shape in response to applied forces, at a continuously applied oscillating shear stress and constant strain [47]. The MRE’s proportionality behavior was slowly reduced as a result of the continuous applied stress at constant strain. Larger scale localization at the molecular level was developed at a wide range of test duration (84,000 s), and softening behavior to the MRE was further instigated. As a result, as shown in Figure 6, the storage modulus was observed to decrease near the end of the durability cycles.

The graph in Figure 6 indicates that the storage modulus behavior’s comparability correlates to the durability of stress relaxation test duration under the magnetic influence. At 1 A and 2 A of applied current, as in Figure 6a, the most significant changes in storage modulus value can be observed. In the case of 1 A applied current, the initial shear storage modulus decreased by approximately 13%, while the initiation value decreased by 11% for 2 A applied current. At this stage, the matrix’s elasticity is ideal for microparticles’ interaction with magnetic flux, with good reversible and instantaneous return capability. The deformation energy was lost as a result of the smallest broken free movable bridge fragment across the microplasticity, which is no longer incorporated into the molecular network. In contrast to the off-state situation, the obvious transformation on storage modulus was deemed strong. The previous research [48] briefly addressed and documented the off-state storage modulus behavior under durability. In comparison, at higher magnetic inducement, the storage modulus changed just slightly. At high fields, the materials may become magnetically saturated. According to the previously available vibrating-sample magnetometer (VSM) results, the CIP used in this study has an average magnetization saturation (M_s_) of 137.06 emu/g. As a result, with known density, the CIP has maximum saturation at around 1.03 T. As a result, increasing the magnetic induce current from 3 A, 4 A, and up to 5 A results in a smaller increment of the initial shear storage modulus value, as shown in Figure 6b. The smallest decrement obtained was 0.7%, and the pattern is consistent throughout the test, especially in the vicinity of the LVE limits.

The strain of 0.01% used for the stress relaxation test was found to be a good predictor of the material response to magnetic stimulation, as it agrees well with the theory that elasticity is dominant. This was highlighted as a flawless attempt at stress relaxation testing for MRE that is nearest to its state of rest and follows Newton’s first rule or is considered a condition of equilibrium. The indicated area represents the beginning of solid/gel transition state behavior with a phase shift angle of about 45°. The apparent shift in storage modules for the entire condition, on the other hand, can be observed mostly beyond the LVE limits. The gradual breakdown of the MRE’s superstructure causes an apparent downtrend in the non-linear area after 1% shear strain. The viscous component has a phase shift angle of more than 45° to 90°, and energy is usually dissipated rather than retained.

Prior to the durability test, CIP appears to be evenly distributed across the silicone rubber and varies in size. Figure 7 depicts the FESEM image of samples containing randomly dispersed particles that agrees with the FESEM images in Figure 3. Figure 7b, however, demonstrates the distinct feature of a shear band that has undergone stress relaxation. As more slippages at molecular structure level and cross-link losses occur in the MRE, the molecular structure becomes insufficiently reconfigurable, resulting in a permanent deformation (microplasticity or shear band) [47]. The image was taken from a similar MRE sample of 70 wt% CIP that was tested, and the resulting value is plotted in the graph in Figure 6. The findings revealed a link between stress relaxation and the MRE life cycle. The test parameters and MRE physical factors have both contributed to the storage modulus decrement over the specified test duration. In comparison with the previous stress relaxation studies on MRE [49,50], this relationship between microstructural failure and the storage modulus characteristic is currently the only established finding. The relaxation of stress in the MRE’s molecular structure is one possible explanation for the storage modulus and durability correlation. In Figure 7c, the presence of shear bands of varying size, thickness, pattern, and location will be the most likely evidence as the factor for storage modulus decreased after durability.

## 4. Conclusions

This analysis and investigation resulted in the characterization of the MRE ‘in phase’ portion. The elastic response and stored energy of the specified MRE corresponding to the applied magnetic field are determined by storage modulus performance. Identification and verification of the LVE region, in relation to the microparticles fraction, play a role in determining the best possible composition for an even more significant study. The proposed MRE behavior model with different weight percent fractions represents the characteristic of storing energy performance towards the potency elasticity limit. The various storage capacity behavior, as well as the relationship to microparticles fraction and alignment, has opened up tremendous potential for MRE in specific desired applications. For instance, owing to time and cost constraints, durability performance investigation within elastic regions necessitates a specific constitution in order to obtain accurate and practical results, which would otherwise be ineffective. The efficacy of 70 wt% CIP isotropic MRE was demonstrated, with a good association between the characteristics of before and after stress relaxation durability assessment. As a consequence, the findings of this study could be useful in future investigations of MRE durability properties with various parameter circumstances. The specific future investigations include the formulation of the mathematical equation, which can describe model of MRE behavior with different wt% microparticles fraction, and the selection of new raw material ranges to achieve a novel type of MRE that can withstand even greater durability challenges.

## Figures and Tables

**Figure 1 micromachines-12-00948-f001:**
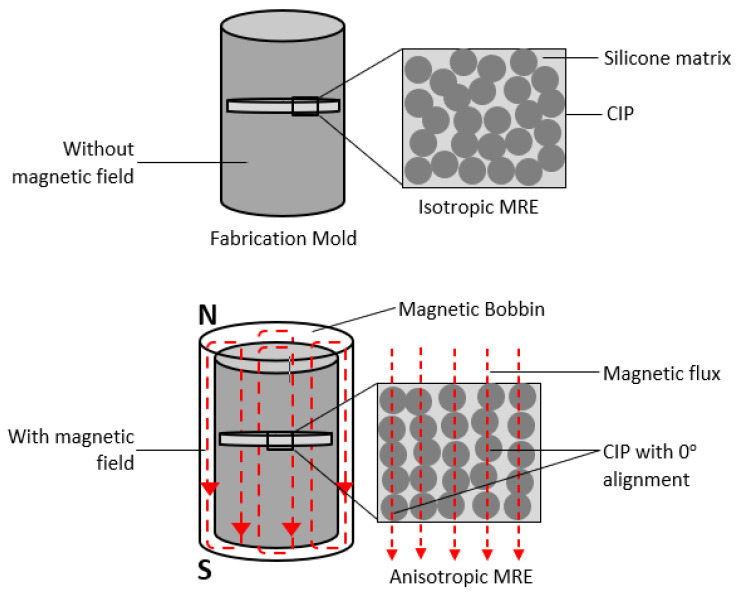
Diagram of isotropic and anisotropic MRE sample preparation.

**Figure 2 micromachines-12-00948-f002:**
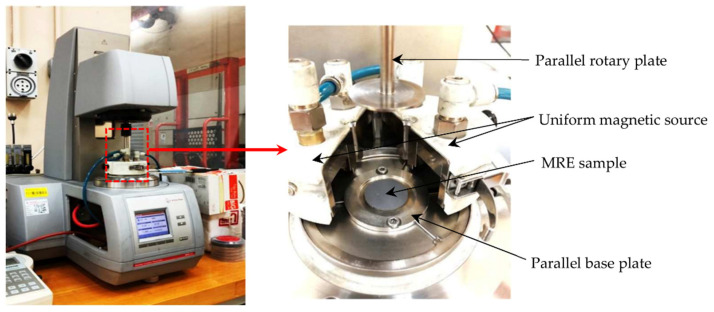
MCR 302 rheometer, MRD 70/1T magnetorheological device, and sample on a base plate.

**Figure 3 micromachines-12-00948-f003:**
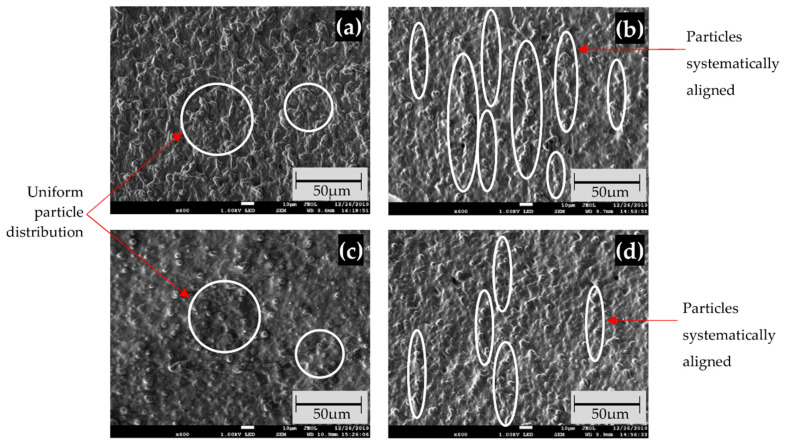
FESEM images at ×600 magnification of particle dispersion in (**a**) high wt% isotropic distribution; (**b**) high wt% anisotropic distribution; (**c**) low wt% isotropic distribution; and (**d**) low wt% anisotropic distribution. The inserted scale bar is 1:5 of FESEM original dimension.

**Figure 4 micromachines-12-00948-f004:**
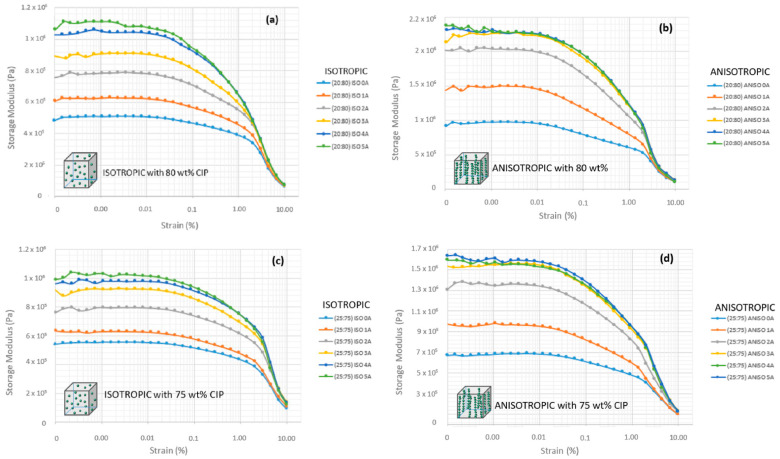
The storage modulus behavior at various microparticles fraction and alignment of (**a**) isotropic (ISO) sample at 80 wt% CIP; (**b**) anisotropic (ANISO) sample at 80 wt% CIP; (**c**) isotropic sample at 75 wt% CIP; (**d**) anisotropic sample at 75 wt% CIP; (**e**) isotropic sample at 70 wt% CIP; (**f**) anisotropic sample at 70 wt% CIP; (**g**) isotropic sample at 60 wt% CIP; (**h**) anisotropic sample at 60 wt% CIP; (**i**) isotropic sample at 50 wt% CIP; and (**j**) anisotropic sample at 50 wt% CIP.

**Figure 5 micromachines-12-00948-f005:**
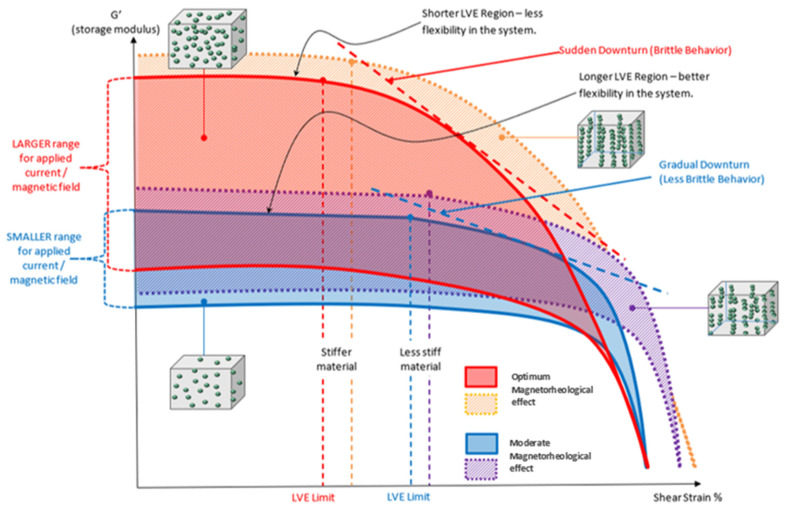
Model of MRE behavior with different wt% microparticles’ fraction at steady state condition.

**Figure 6 micromachines-12-00948-f006:**
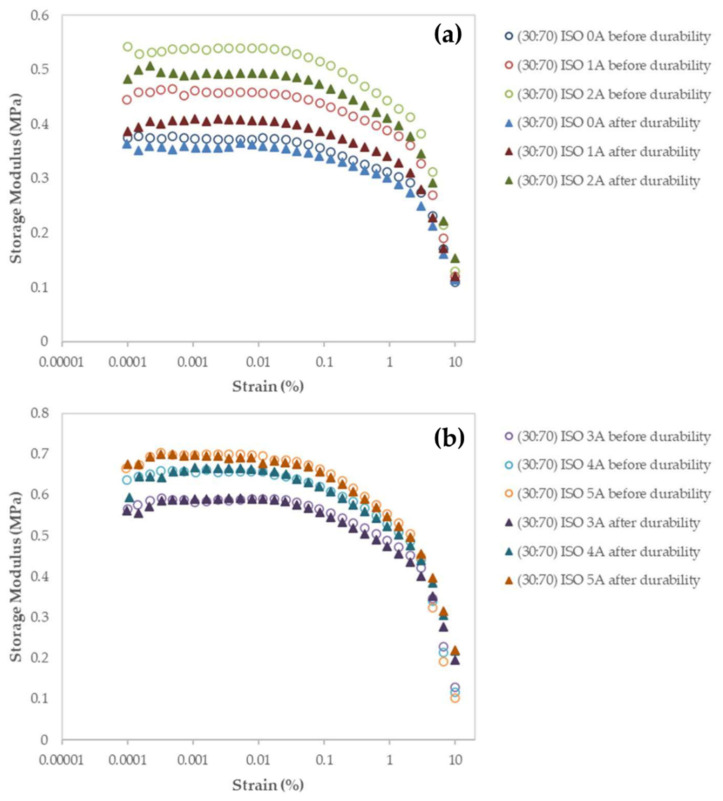
Storage modulus of isotropic (ISO) MRE with 70 wt% CIP under durability evaluation: (**a**) storage modulus behavior at 0 A, 1 A, and 2 A; (**b**) storage modulus behavior at 3 A, 4 A, and 5 A.

**Figure 7 micromachines-12-00948-f007:**
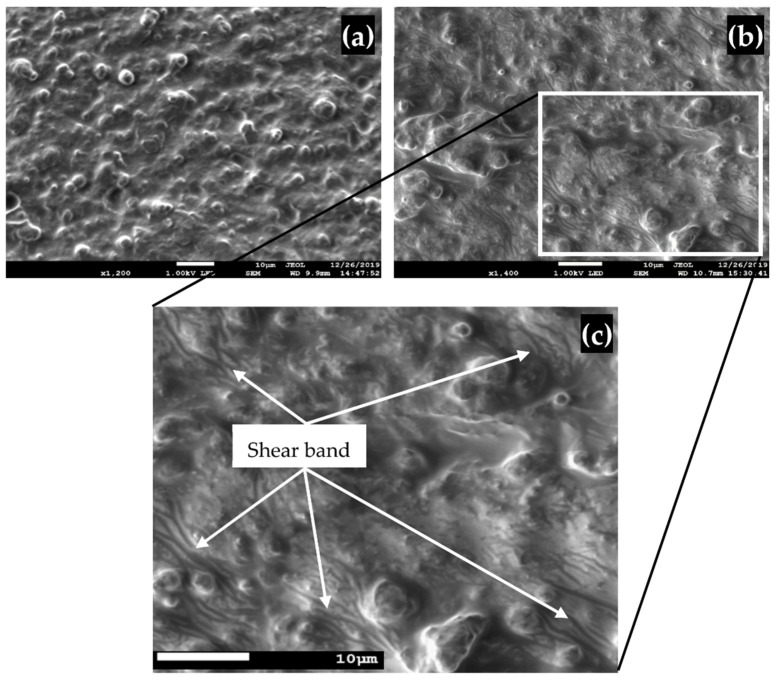
FESEM images of MRE samples before and after durability test: (**a**) MRE before durability evaluation; (**b**) sample after durability by stress relaxation; and (**c**) view of the shear band deformation in the MRE matrix in greater detail.

## Data Availability

The raw/processed data required to reproduce these findings cannot be shared at this time as the data also form part of an ongoing study. In future, however, the raw data required to reproduce these findings will be available from the corresponding authors.

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
