# Peer review of "The Effect of Microparticles on the Storage Modulus and Durability Behavior of Magnetorheological Elastomer"

_micromachines, 2021, doi:10.3390/mi12080948_

Round 1

Reviewer 1 Report

The study by Johari M.A.F. et al. investigates the effect of particle content on storage modulus and relaxation behavior of silicone-based MREs. The effect of the particle distribution is studied as well. Although such MREs have been a subject of numerous studies, this work has clear objectives, and gives new insight into the topic by finding new correlations between the referred properties. Probably the most important result is the model scheme generalizing strain behavior of the MREs with various structure. However, there are several remarks that should be addressed:

Introduction:

1) Line 46: To demonstrate a broad impact of the MREs, their applications in different fields should be explicitly addressed, such as: electronics and sensing components (10.1016/j.ijmecsci.2020.105816), microfluidics (10.1088/0964-1726/25/2/025011), military and radio-absorbing devices (10.1088/1361-665X/aa7ef6) etc.

2) Line 62: Besides the mentioned particle parameters (size, shape), there is also evidence that particle surface characteristics, such as coating (10.3390/polym10121411), play a significant role in tuning the MREs properties.

3) The authors performed durability testing of the MREs. By a definition, durability testing is referred to a fatigue or endurance testing, usually long term. However, herein, the authors used stress relaxation technique. Could you please briefly specify, why have you selected this type of durability evaluation?

Methodology:

4) The curing of the MREs was performed at 25°C (line 104), which is quite unusual considering a short curing time (of 2 hours). The authors claim that no evidence of sedimentation was observed. The curing kinetics of the matrix would be helpful to support the claims above.

5) Line 121: What was the initial clamping force in the rheometer?

6) Line 151: Could you please specify testing time period also in different units, e.g. hours

Results and discussion:

7) Figure 1: the scale bar is too small; it is difficult to see or guess the dimensions

8) Figure 2: artwork should be improved, a clearer presentation of results should be required, legend is vague (particularly, Figure 2d). For a clearer presentation of the observed trends, is it possible to plot small-strain G’ against magnetic field strength for all the samples?

9) Figure 3: again, small descriptions, hard to see in details

10) Line 265: the claims about the shear band regions should be supported by the original literature source

11) Line 287-289: “…Magnetically, the materials appear to be saturated at high fields. As a result, increasing the magnetic induce current from 3A, 4A, and up to 5A results in a smaller increment of the initial shear storage modulus value…” To demonstrate the level of material saturation, the true magnetic fields (instead of coil currents) should be correlated to the VSM curves of the MREs (or at least, VSM of CI powder).

12) Only half of the samples was subjected to stress relaxation durability testing. The study would be more impactful, if also anisotropic samples would be investigated for their durability.

Also, the attention should be paid to language, some misprints appeared here (line 44, 140, 176, 185, 334…) etc. Please revise.

Reviewer 2 Report

  1. The paper does not specify which source of the magnetic field was used to fabricate the MAE with an anisotropic structure. It is indicated that it was parallel to the thickness of the mold. This aspect is very important, since the structure of the final material depends on the orientation and parameters of the magnetic field. At a thickness of 50 mm, an inhomogeneous distribution of the magnetic field induction and differences in the anisotropic structure over the sample thickness are inevitable.

Separately, it is necessary to explain how the trajectories of the closure of the field lines of the magnetic field source are located. Perhaps the lines, passing through the sample, are closed in an arc, and not in a straight line, as the authors try to show. In this case, the sample preparation was carried out incorrectly, which reflects the validity of all the conclusions obtained.

  1. Visual assessment of the samples in Fig. 1. raises questions:

- images differ in light intensity: anisotropic samples are lighter. Perhaps this was done on purpose to simplify the selection of material structuring zones. However, if you remove the marks highlighted by the authors, it is not possible to find the difference between an isotropic and anisotropic sample.

- Figure b and d clearly show striae (strands) in the material matrix. This may be evidence of a change in the position of the magnetic field source during sample polymerization.

All this refers us to clarification on the first question: how was the source of the magnetic field located?

  1. Fig. 1, fig. 2 exclude the possibility of reading them and analyzing the information. Corrections required.
  2. It is known that composite polymers with solid non-deformable fillers at a concentration above 80% lose stability and demonstrate the effect of inverse dispersion (Bondaletova L.I. B811 Polymer composite materials (part 1): tutorial / L.I. Bondaletova, V.G. Bondaletov - Tomsk: Publishing house of the Tomsk Polytechnic University, 2013. - 118 p.). These polymers are physically difficult to manufacture due to their instability. The process of their manufacture, as well as the planned area of application, requires clarification.
  3. The work does not include schemes for conducting experiments, in particular when conducting experiments using a magnetic field. The reliability of the results largely depends on its orientation.
  4. line 159. Are you talking about "agglomeration" or "structural organization"? The degree of agglomeration is not visually traced and the authors do not analyze it. Filler clumping may be due to mixing technology (data not available) during manufacture and curing. And in any case, it logically increases with the growth of the filler content.
  5. line 189. What does “optimal magnetization” mean? Is it about magnetic saturation of the material?
  6. It makes sense to equalize the scales in Fig. 2.
  7. line 202 - What criteria of optimality were used by the authors?
  8. line 234 - A well-known fact.
  9. line 334. These lines are similar to those shown in fig. 1 b and d. The figure does not indicate whether it is an isotropic or anisotropic sample. It may also be due to the peculiarities of sample preparation (mixing, polymerization, cutting). The authors refer to these bands as confirmation of the results. Note that in this case, the filler should move, forming grooves and "wound channels". But the main thing is that the filler-matrix contact must be destroyed. This is obviously not observed in the figures.
  10. Authors should clarify the terminology used.
  11. In the literature with which the researchers worked, and which is referred to in confirming the results obtained, there are many works of the category of "self-citation". This may be the reason for an insufficient in-depth critical analysis of the results obtained and not paying attention to the peculiarities of material manufacture.
  12. We recommend that authors pay attention to the following works:

- Çakmak, U.D.; Fischlschweiger, M.; Graz, I.; Major, Z. Adherence Kinetics of a PDMS Gripper with Inherent Surface Tackiness. Polymers 202012, 2440. https://doi.org/10.3390/polym12112440 ;

- Vasilyeva, M.; Nagornov, D.; Orlov, G. Research on Dynamic and Mechanical Properties of Magnetoactive Elastomers with High Permeability Magnetic Filling Agent at Complex Magneto-Temperature Exposure. Materials 202114, 2376. https://doi.org/10.3390/ma14092376

- Shuib, R.K.; Pickering, K.L.; Mace, B.R. Dynamic properties of magnetorheological elastomers based on iron sand and natural rubber. J. Appl. Polym. Sci. 2015132, 41506.

- Hiptmair, F.; Major, Z.; Hasslacher, R.; Hild, S. Design and application of permanent magnet flux sources for mechanical testing of magnetoactive elastomers at variable field directions. Rev. Scient. Instr. 201586, 085107.

- Lu, X.; Qiao, X.; Watanabe, H.; Gong, X.; Yang, T.; Li, W.; Sun, K.; Li, M.; Yang, K.; Xie, H.; et al. Mechanical and structural investigation of isotropic and anisotropic thermoplastic magnetorheological elastomer composites based on poly(styrene-b-ethylene-co-butylene-b-styrene) (SEBS). Rheol. Acta 201151, 37–50. 

The work is written on a relevant topic. However, it needs significant improvement. At this stage, there is no clearly defined scientific novelty in the work. It is also very abstractly suggested that the results obtained are of practical value. This should be described in more detail.

Round 2

Reviewer 2 Report

The authors showed a serious approach to the analysis of the reviewer's comments.

Most of the questions are fully answered.
However, in the opinion of the reviewer, some questions remained unresolved:

1. The main issue is the use of the term "algorithms". "Agglomeration otherwise aggregation is a spatial grouping and adhesion of dispersed particles, as a result of which larger" secondary "particles are formed."

Agglomeration can be discussed if spatial structures of the "sphere in sphere" type are considered. In one of the works to which the authors refer, this issue is considered. In this case, the use of "structure organization" is justified.

2. Unfortunately, the correction made in the article did not allow us to clarify the issue of optimality. The authors write "CIP reached its optimal level magnetization at a higher applied current ...". However, the question arises as optimal for what? Or is it still the "required level"?
